# Circulating Mesenchymal Stromal Cells in Patients with Infantile Hemangioma: Evaluation of Their Functional Capacity and Gene Expression Profile

**DOI:** 10.3390/cells13030254

**Published:** 2024-01-29

**Authors:** Carlotta Abbà, Stefania Croce, Chiara Valsecchi, Elisa Lenta, Rita Campanelli, Alessia C. Codazzi, Valeria Brazzelli, Adriana Carolei, Paolo Catarsi, Gloria Acquafredda, Antonia Apicella, Laura Caliogna, Micaela Berni, Savina Mannarino, Maria A. Avanzini, Vittorio Rosti, Margherita Massa

**Affiliations:** 1General Medicine 2—Center for Systemic Amyloidosis and High-Complexity Diseases, IRCCS Policlinico San Matteo Foundation, 27100 Pavia, Italy; c.abba@smatteo.pv.it; 2Immunology and Transplantation Laboratory, Cell Factory, Pediatric Haematology Oncology, IRCCS Policlinico San Matteo Foundation, 27100 Pavia, Italy; s.croce@smatteo.pv.it (S.C.); c.valsecchi@smatteo.pv.it (C.V.); e.lenta@smatteo.pv.it (E.L.); g.acquafredda@smatteo.pv.it (G.A.); ma.avanzini@smatteo.pv.it (M.A.A.); 3Center for the Study of Myelofibrosis, IRCCS Policlinico San Matteo Foundation, 27100 Pavia, Italy; r.campanelli@smatteo.pv.it (R.C.); a.carolei@smatteo.pv.it (A.C.); p.catarsi@smatteo.pv.it (P.C.); v.rosti@smatteo.pv.it (V.R.); 4Pediatric Cardiology, IRCCS Policlinico San Matteo Foundation, 27100 Pavia, Italy; a.codazzi@smatteo.pv.it (A.C.C.); anto.apicella@smatteo.pv.it (A.A.); 5Institute of Dermatology, IRCCS Policlinico San Matteo Foundation, 27100 Pavia, Italy; v.brazzelli@smatteo.pv.it; 6Orthopedics and Traumatology Clinic, IRCCS Policlinico San Matteo Foundation, 27100 Pavia, Italy; l.caliogna@smatteo.pv.it (L.C.); micaela.berni@hotmail.com (M.B.); 7Pediatric Cardiology Unit, V. Buzzi Children’s Hospital, 20154 Milan, Italy; savina.mannarino@asst-fbf-sacco.it

**Keywords:** infantile hemangioma, mesenchymal stromal cells, propranolol, angiogenesis

## Abstract

We previously published that in patients with infantile hemangioma (IH) at the onset (T0) colony forming unit-fibroblasts (CFU-Fs) are present in in vitro cultures from PB. Herein, we characterize these CFU-Fs and investigate their potential role in IH pathogenesis, before and after propranolol therapy. The CFU-F phenotype (by flow cytometry), their differentiation capacity and ability to support angiogenesis (by in vitro cultures) and their gene expression (by RT-PCR) were evaluated. We found that CFU-Fs are actual circulating MSCs (cMSCs). In patients at T0, cMSCs had reduced adipogenic potential, supported the formation of tube-like structures in vitro and showed either inflammatory *(IL1β* and *ESM1*) or angiogenic (*F3*) gene expression higher than that of cMSCs from CTRLs. In patients receiving one-year propranolol therapy, the cMSC differentiation in adipocytes improved, while their support in in vitro tube-like formation was lost; no difference was found between patient and CTRL cMSC gene expressions. In conclusion, in patients with IH at T0 the cMSC reduced adipogenic potential, their support in angiogenic activity and the inflammatory/angiogenic gene expression may fuel the tumor growth. One-year propranolol therapy modifies this picture, suggesting cMSCs as one of the drug targets.

## 1. Introduction

Infantile hemangioma (IH) is a vascular tumor that spontaneously declines [1], although 10–12% of subjects present complications [2]. In addition to mature endothelial cells (ECs), heterogeneous subsets of cells have been identified within the tumor mass, such as endothelial progenitor cells (EPCs), mesenchymal stromal cells (MSCs) and pericytes [3,4,5,6]. The involution phase of IH is characterized by the appearance of loose fibrofatty tissue and it has been hypothesized that one of the mechanisms that contributes to the adipogenesis involves hemangioma-derived MSCs exhibiting multi-lineage differentiation with robust adipogenic potential [7] residing in the perivascular region of the tumor [8]. In addition, Khan et al. reported the presence of multipotential stem cells that give rise to hemangioma-like lesions in an immune deficient mouse model, in contrast to the long-held view that IH arises from ECs [9].

Propranolol, a β-adrenergic receptor antagonist, has become a valuable treatment for severe IH, and has been the first FDA-approved medication for this disease [10]. The mechanism underlying the benefit of this drug is still under investigation [11]. A recent in vitro study suggests a role for propranolol in accelerating the trans-differentiation of hemangioma-derived stem cells into adipocytes, therefore preventing the growth of proliferating IH [12] and suggesting that the identification of cell differentiation modulators in hemangioma may provide a tool for favoring involution or interfering with cellular differentiation processes.

In a previous study [13], we investigated the presence of circulating EPCs in infants with IH at the onset (T0) and during one year therapy with propranolol. Our data indicated that circulating EPCs play a role in IH pathogenesis and that, during the therapy with propranolol, there is an increased frequency of circulating endothelial colony forming cells (ECFCs) that may be due to either the involution of the tumor mass or a decrease in its ECFC recruitment function. In in vitro culture experiments performed to grow ECFCs we incidentally observed colony forming unit-fibroblast cells (CFU-Fs) that we isolated and stored in liquid nitrogen. This finding has never been reported by other groups, and it has never been evidenced in our previous studies on ECFCs in different pathologies of both adults [14,15] and infants [16]. Recent data in the literature indicate that MSCs can be detected in the peripheral blood (PB) of patients with cardiovascular pathology [17], multiple sclerosis [18] and other non-physiological conditions (Extra Corporeal Membrane Oxygenation) [19] by flow cytometry. In addition, some studies have reported circulating MSCs (cMSCs) in healthy subjects [20,21]. Herein, we have studied the previously stored CFU-Fs from patients with IH at T0 and during therapy with propranolol. We assessed their capacity to differentiate into different cell lineages and their possible role in fueling angiogenic processes of IH. We analyzed the transcript expression of genes in CFU-Fs from patients with IH and healthy subjects comparable in age (CTRLs), but also in MSCs derived from different sources such as bone marrow (BM) and umbilical cord tissue (UCt).

## 2. Materials and Methods

### 2.1. Patients and Clinical Procedures

We took advantage of the 18 patients with IH, showing eligibility criteria for systemic oral treatment with propranolol, enrolled in a previous study [13] at T0 (median age 6 months, range 2–11) and re-evaluated at 3, 6, 9 and 12 months during the therapy with propranolol; in the same study 24 CTRLs (median age 7 months, range 1–30) were also enrolled. Propranolol was administered orally to 17 patients (2 mg/kg/day starting from a dosage of 0.5 to 1 mg/kg/day), one patient was enrolled in the study but the parents refused to start the therapy after the blood sample for preliminary evaluations was obtained [13]. In the present study we included 8/18 of those patients and 2/24 CTRLs who had CFU-F colonies growing in in vitro cultures performed to obtain PB ECFCs at the onset of the disease. Where indicated, we performed comparisons with BM-derived MSCs of 5 healthy children (median age 39 months, range 24–60 months) who came to our institution as BM hematopoietic stem cell donors; similarly, stored UCt-derived MSCs (*n* = 7) were used [22].

### 2.2. In Vitro Cultures Generating CFU-F Cells

Mononuclear cells (MNCs) obtained from 2–3 mL of heparinized PB of patients with IH or CTRLs (the leftover of blood samples taken for clinical purposes) were seeded in a culture dish in the presence of EGM-2MV complete medium; from day 10 to day 20, by light microscopy, co-existing ECFCs and CFU-Fs were observed [13]. CFU-F colonies were isolated by cloning cylinders and re-plated as previously reported [23].

Medium was changed twice a week until the cells reached the confluence ≥ 80%. At this time, cells were detached by recombinant trypsin-EDTA (Euroclone, Pero, Italy) and re-plated for expansion until senescence. On the other hand, ECFCs were trypsinized and expanded for 2–3 passages in vitro to obtain a number of ECFCs sufficient for further experiments [16]. Both the CFU-Fs and the ECFCs have been characterized to confirm their belonging to either the mesenchymal stromal or endothelial cell lineage (see Section 3). We also tested human umbilical vein endothelial cells (HUVECs) as positive control of the endothelial counterpart.

Human UCts were collected from healthy pregnant women at full term who underwent elective cesarean section at the Department of Obstetrics and Gynecology of the IRCCS Policlinico San Matteo Foundation. The UCts were cut into fragments and incubated with collagenase type II 1 mg/mL (Sigma-Aldrich, St. Louis, MO, USA) at 37 °C in a humidified 5% CO_2_ atmosphere for 30 min. Digestion was continued for an additional 30 min with trypsin-EDTA 1 U/mL (Lonza, Basel, Switzerland). Cell suspension was collected, and MNCs were isolated by density gradient centrifugation and plated in non-coated polystyrene culture flasks (Corning Costar, Corning, NY, USA) at a density of 160,000 cells/cm^2^ in complete culture D-MEM (Gibco LifeTechnologies Limited, Paisley, UK).

### 2.3. Cell Phenotypes

One ×105 CFU-F-derived cells were incubated with 5 µL of FITC or PE-anti-human-CD34, CD45, CD73, CD90, CD31, CD105 and class I-HLA monoclonal antibodies and appropriate isotype (all from Beckman Coulter, Milan, Italy). Cell acquisition was performed by Navios flow cytometer (Beckman Coulter) and analysis was performed by the Kaluza software, 2.1 version (Beckman Coulter). Similarly, 1 × 105 ECFC-derived cells were incubated with anti-human CD45, CD34, CD146, CD31, CD144, VEGFR2 and CD105 (all from BD, Pharmingen, San Diego, CA, USA). The percentage of positive cells was calculated, subtracting the value of the appropriate isotype controls.

### 2.4. Proliferative Capacity

The proliferative capacity of the CFU-F cells was expressed as cumulative population doubling (cPD) where the PD was expressed by the formula:Log (number of harvested cells/number of plated cells)/log_2_ at each passage.

### 2.5. Differentiation Capacity

The differentiation capacity of CFU-Fs was assessed at P2–P4 as previously described [24]. Briefly, to induce osteogenic differentiation, adherent cells were incubated in α-minimal essential medium (Euroclone), 10% FBS (Euroclone), 50 µg/mL penicillin, 50 mg/mL streptomycin and 2 mmol/L L-glutamine, supplemented with 10^7^ mol/L dexamethasone, 50 mg/mL-ascorbic acid and 5 mmol/L glycerol phosphate (all from Sigma-Aldrich).

To induce adipogenic differentiation, the medium described above was supplemented with 100 mg/mL insulin, 50 mmol/L isobutyl methylxanthine and 0.5 mmol/L indomethacin (all from Sigma-Aldrich). For chondrogenic differentiation, the cells were cultured in DMEM F12-HAM containing 1% FBS, 100 nM dexamethasone, 0.05 mM ascorbic acid, 10 ng/mL transforming growth factor-β1, 1% insulin-transferrin-sodium selenite (Sigma-Aldrich). Extracellular matrix protein accumulation was stained with 1% Alcian blue 8GX in 3% acetic acid, pH 2.5 (Bio-Optica, Milan, Italy).

### 2.6. Gene Expression Analysis

Total RNA was extracted by the miRNeasy Mini Kit (Qiagen, Hilden, Germany). The cDNA synthesis was performed using the iScript Kit (Bio-Rad, Hercules, CA, USA) and analyzed using the qPCR method. Predesigned PrimePCR SYBR Green assays were purchased from Bio-Rad. Quantification of transcripts was performed in 20 μL reaction mix with 1× SsoAdvanced Universal SYBR Green SuperMix (Bio-Rad). Primers pairs were lyophilized in the wells of the PCR plates. PCR conditions were: 95 °C for 30 s followed by 40 cycles at 95 °C for 5 s and at 58 °C for 5 s. Melting curves were generated after amplification in the range 65–95 °C with increments of 0.2 °C every 10 s. For each experiment, 4 μL of cDNA, corresponding to 10 ng of total RNA, were used. The PCR data were collected using CFX96 RealTime System (Bio-Rad). Each sample was tested in triplicate. Calculation of normalized relative expression levels was performed using the ΔΔCq method. Normalization was performed using stable reference genes (*GAPDH*, *YWHAZ*, *UBC*, *HPRT1*, *RPLP0*). All analyses were performed with GenEx (version 6.1, MultiD, Goteborg, Sweden).

### 2.7. Peroxisome Proliferator-Activated Receptor Gamma (PPARγ) and CCAAT/Enhancer-Binding Protein Alpha (C/EBPα)

Total RNA from cellular pellets was extracted using the RNeasy Mini Kit (Qiagen) according to the manufacturer’s instructions. RNA was quantified using a NanoDrop (Thermo Fisher Scientific, Waltham, MA, USA). A total of 1 μg of RNA was reverse transcribed into cDNA (Reverse Trascritional M-MLV RT kit; Promega, Milan, Italy). For each experiment, 1 μL of cDNA, corresponding to 100 ng of total RNA, was used. Real-time PCR for *PPARγ* and *C/EBPα* genes was performed on the Real-Time PCR instrument (AB 7500; Applied Bio-systems, Waltham, MA, USA) and data analysis was performed by 7500 fast Real-time PCR systems (Applied Biosystems). Expression levels for each gene were calculated as relative quantification (RQ) using Hypoxanthine phosphoribosyltransferase 1 (*HPRT-1*) as internal control.

### 2.8. In Vitro Angiogenesis Assay

Healthy subject-derived ECFCs (early passages) were resuspended in EGM-2MV complete medium (EBM-2 plus 5% FBS, rhEGF, rhVEGF, rhFGF-B, rhIGF-1, ascorbic acid and heparin). Two ×10^4^ ECFCs/well, in the presence/absence of CFU-Fs (ratio 5:1) were plated in duplicate in 96 well plates coated with a basement membrane extract (Trevigen^®^, Gaithersburg, MD, USA), derived from Engelbreth-Holm-Swarm mouse sarcoma, a tumor rich in extracellular matrix proteins and growth factors. Plates were then incubated at 37 °C, 5% CO_2_. Capillary-like network formation was assessed 4 h later and up to 24 h. In order to assess the ability of cMSCs to favor angiogenesis in vitro, co-culture experiments were also performed in suboptimal conditions, i.e., using endothelial basal medium (EBM-2) supplemented with 2% FCS and without the addition of cytokines.

### 2.9. Statistical Analysis

The frequencies of parametric and nonparametric continuous variables are reported as mean ± standard deviation (SD). Comparison between groups of continuous parametric and nonparametric variables was carried out with the use of Fisher’s exact test for independent samples and the nonparametric Mann–Whitney U test, respectively, applying the Bonferroni correction for multiple tests.

Principal component analysis (PCA) was used to evidence separation of the gene expression between cells of the endothelial compartment (HUVECs, ECFCs) and of the stromal compartment (circulating CFU-Fs, as well as MSCs from different sources). A value of *p* < 0.05 was considered statistically significant for all analyses. All analyses were performed by STATISTICA software (version 8, Dell Technologies Inc., Round Rock, TX, USA).

## 3. Results

### 3.1. CFU-Fs Circulate in PB of Patients with IH at T0

The frequency of patients with IH at T0 showing a PB CFU-F growth in in vitro cultures performed to obtain ECFCs (8/18) was higher (*p* = 0.01) than that of CTRLs (2/24). The CFU-F frequency/10^7^ PB MNCs in patients and CTRLs is shown (Figure 1A). The characteristics of patients at T0 who presented or not CFU-Fs are summarized in Table 1; no significant difference between the two groups of patients was found.

#### 3.1.1. CFU-F Characterization

The CFU-Fs were expanded and then characterized according to the criteria defined by the International Society for Cellular Therapy for MSCs [25]. Briefly, the MSCs must be plastic adherent, show a spindle shape morphology when in culture, express CD105, CD73 and CD90, lack expression of the hematopoietic/endothelial surface molecules, differentiate to osteoblasts, adipocytes and chondroblasts in vitro. According to these criteria, the CFU-Fs under investigation were plastic adherent, displayed the typical morphology (Figure 1B, for one representative patient) and immune phenotype, assessed by flow cytometry at early (P2–P4) and late (P7–P8) passages of MSCs, resulting over 95% positive for CD73, CD105, class I-HLA and CD90 and negative for the hematopoietic/endothelial markers such as CD45, CD31 and CD34 (Figure 1C).

#### 3.1.2. CFU-F Differentiation

The osteogenic and chondrogenic differentiation capacity were present in CFU-Fs isolated from patients with IH and they were comparable to those of CFU-Fs from CTRLs and healthy BM-derived MSCs (Figure 2A–C and Figure 2D–F, respectively). When the adipogenic differentiation capacity was investigated, we found that both the amount and the dimensions of lipid droplets of CFU-Fs from either patients or CTRLs were defective (Figure 3A and Figure 3B, respectively) with respect to those obtained from healthy BM MSCs (Figure 3C); nevertheless, the quality of lipid droplets evidenced in cMSCs was similar to that found evaluating UCt-derived MSCs (Figure 3D) [22,26]. To further investigate the adipogenic process of CFU-Fs, we assessed PPARγ, a transcription factor that stimulates the genetic events that result in MSC adipocyte differentiation, and the C/EBPα, also required for robust adipocyte-specific gene expression. PPARγ expression in adipocytes differentiating from CFU-Fs of patients with IH at T0 was decreased with respect to that of adipocytes differentiating from healthy BM MSCs, but comparable to the PPARγ expression in adipocyte differentiating from either CTRL CFU-Fs or UCt-derived MSCs (Figure 3E). The PPARγ expression in adipocyte differentiating from UCt-derived CFU-Fs was significantly lower (*p* = 0.021) than that evidenced in those differentiating from healthy BM MSCs. Similarly, the expression of C/EBPα in adipocytes differentiating from CFU-Fs of patients with IH at T0 was decreased with respect to that of adipocytes differentiating from healthy BM MSCs, but comparable to that of CFU-Fs from CTRLs and UCts (*p* = 0.03 for UCt versus healthy BM MSCs) (Figure 3F).

In conclusion, PB CFU-Fs of patients with IH have the MSC morphology, phenotype and differentiation capacity in osteocytes and chondrocytes. The defective differentiation in adipocytes with respect to BM MSCs evidences the non-BM source, as indicated by reports from other groups [27]. Altogether, these data allow us to define PB CFU-Fs as cMSCs.

**Table 1 cells-13-00254-t001:** Characteristics of patients with IH at the onset showing CFU-Fs (yes CFU-Fs) or not (no CFU-Fs) in in vitro cultures.

	Yes CFU-Fs (*n* = 8)	No CFU-Fs (*n* = 10)
Sex (male/total)	4/8	6/10
Gestational age (wk)	38 ± 3	36 ± 2
Weight (g)	2775 ± 847	2005 ± 471
Delivery (Caesarean section/total)	4/8	6/10
Maternal pathology (*n*/total)	2/8	1/10
Antenatal prednisone (*n*/total)	4/8	6/10
Cardiac pathologies (*n*/total)	1/8	2/10
Hemangioma dimension at birth: <5 mm	6	6
>5 mm	2	4
Hemangioma number at sampling: 1	6	7
>1	2	3
IH aspect at birth: Surface	7	9
Deep	2	2
Mixed	0	0
Position: Head and neck	6	5
Thorax abdomen	2	4
Upper and lower limbs	1	4
Glutes	1	2
Hemangioma dimension at sampling: <2 cm	1	0
>2 cm	7	10
Colour: Light	3	1
Strong	5	9
Complication (*n*/total)	0/8	3/10
Complete regression *	7/7	6/10

* The complete regression has been defined according with the score index recently described in Volontè et al. [28].

### 3.2. cMSCs in Patients with IH Receiving Propranolol

The eight patients presenting cMSCs at T0 received propranolol for one year. The study performed in all the available patients at 3, 6, 9 and 12 months of therapy, showed a decrease of the frequency of CFU-Fs/10^7^ MNC (Figure 4A).

Assessed as cPD, the cMSC proliferative capacity was comparable among patients at T0 (*n* = 8), patients receiving therapy since 9–12 months (T9 *n* = 3 and T12 *n* = 3) and CTRLs (*n* = 2) (Figure 4B).

We observed that cMSCs from patients assessed at T0 or during therapy with propranolol for 9–12 months, and from donor BM MSCs, entered senescence at comparable passages (median P12, P11 and P12, respectively) (Figure 4C). The cMSCs from CTRLs (*n* = 2) entered senescence at passages 5 and 16, respectively, suggesting a distribution comparable to that observed in patient cMSCs and healthy BM MSCs.

cMSC adipogenic differentiation capacity was assessed during propranolol therapy (T9–T12) (*n* = 3) and compared to that at T0 (*n* = 3) by a transversal study assessing the expression of PPARγ and C/EBPα genes. PPARγ and the C/EBPα gene expression were increased during the treatment with propranolol with respect to T0 (Figure 4D); supporting this finding, we observed the morphological appearance of lipid droplets in cMSCs from patients at T9–T12 of therapy with propranolol cultured in medium without the stimuli that induce the adipogenic differentiation, but not in cMSCs from patients at T0 (Figure 4E and Figure 4F, respectively, for one representative patient).

### 3.3. Role of cMSCs in the In Vitro Angiogenesis Assay

To investigate a possible role of cMSCs in the angiogenic process that characterizes the IH, we performed an in vitro tube formation assay based on the co-culture of ECFCs from healthy subjects, in the presence of cMSCs from patients with IH at T0 (*n* = 4), or from CTRLs (*n* = 2); in addition, we assessed the co-culture of ECFCs and donor BM MSCs (*n* = 5). The co-cultures were performed in suboptimal conditions. The results in 3/3 experiments showed that healthy ECFCs were able to form tube-like structures in standard conditions (Figure 5A), but not in suboptimal conditions (Figure 5B). When ECFCs were co-cultured in suboptimal conditions with cMSCs from patients with IH at T0, tube-like structures were observed (Figure 5C), while no tube-like structures were present in co-culture of ECFCs with cMSCs of CTRLs (Figure 5D) or MSCs from BM donors (Figure 5E). When ECFCs were co-cultured with cMSCs from patients with IH receiving propranolol for 12 months (*n* = 3), no tube-like structures were evidenced (Figure 5F). We could not perform the experiment using cMSCs from a CTRL comparable in age since, although 7/24 previously enrolled CTRLs were more than 12 month old, none of them formed CFU-Fs. No tube-like structures were ever seen when we cultured IH or CTRL cMSCs, or BM MSCs in the absence of ECFCs.

### 3.4. cMSC Gene Expression Analysis

cMSCs were identified during in vitro cultures performed to obtain ECFCs, therefore, we first wanted to rule out any relation between MSCs and the endothelial lineage at a molecular level. We assessed the expression of 147 genes (Appendix A) associated to the angiogenic process in a number of different cell types: cMSCs from patients with IH at T0 (*n* = 3), cMSCs from CTRLs (*n* = 2), BM MSCs from healthy adults (*n* = 3), HUVECs (*n* = 2) and ECFCs from CTRLs (*n* = 2). The PCA evidenced a clear separation of the gene expression between cells of the endothelial compartment (HUVECs, ECFCs) and MSCs; no gene expression subtype cluster was present within the cell populations (Figure 6A). To investigate thoroughly the cMSC gene expression of patients with IH, we restricted the analysis to 48 genes that were selected according to their differential expression in the initial analysis of 147 genes (Appendix A highlighted by boldface and Appendix A). The PCA of cMSCs from patients and CTRLs versus BM MSCs showed a clear gene expression difference (Figure 6B and Appendix A). The gene expression analysis of cMSCs of IH at T0 (*n* = 5) compared to cMSCs of CTRLs (*n* = 2) showed a significant overexpression of *IL1*β (*p* = 0.046) and of *F3* (*p* = 0.019) genes, and a downregulated expression of the *TGF*α (*p* = 0.034) and *ESM1* (*p* = 0.048) genes (Figure 7). To investigate the gene expression of cMSCs from IH patients following propranolol treatment, we evaluated patients with IH (*n* = 5) at T0 and after 9–12 month therapy with propranolol (T9–T12) (*n* = 4). As shown (Figure 7), cMSCs from patients receiving therapy showed a lower expression of *F3* (*p* = 0.035) and a higher expression of *TGF*α (*p* = 0.029) and *EDN1* genes (*p* = 0.0088) with respect to patients at T0. No significant difference was found between the gene expression of cMSCs from T9 to T12 patients with IH and cMSCs from CTRLs.

## 4. Discussion

This study shows the presence of cMSCs in 44% of the patients with IH at T0 and in 8% of CTRLs. These cells, isolated by in vitro cultures aimed to expand circulating ECFCs, have been described neither in the literature nor in our previous studies [14,15]. They had a phenotype and an osteogenic/chondrogenic differentiation capacity comparable to that of BM MSCs, while the differentiation in adipocytes was defective with respect to the BM MSCs. It has been reported that MSCs derived from involuting IH form a lower number of adipocytes/amount of lipids during adipogenesis than those derived from proliferative IH, but comparable to that of tissue-derived MSCs (i.e., normal infant foreskin) [7]. This finding allows us to speculate about the origin of cMSCs. Indeed, although it has been shown that cancer cells release specific factors that induce MSC mobilization and recruitment to the tumor stroma [29], it is unlikely that cMSCs come from the BM because of the significantly different expression profile. On the other hand, MSCs in IH reside in the perivascular region and may migrate to the PB; however, the tumor mass of patients at the onset and out of therapy has been described in the proliferative phase, characterized by stromal cells with a high adipogenic potential [8]. Finally, the possibility exists that cMSCs in patients with IH derive from other, not yet identified, normal tissues, and modify their adipogenic capacity once in the tumor. It is difficult to verify this task because the successful administration of propranolol has made surgical intervention in young patients extremely rare, preventing the availability of hemangioma tissue and the study of resident MSCs.

The presence of cMSCs in a quite high percentage of patients with IH may also be related to their age (<12 months). Early post-natal life is associated with the transient presence of circulating progenitor cells [30]. However, our previous evaluation of circulating ECFCs in 96 preterm infants, by the same in vitro culture, never showed co-existing cMSCs, therefore indicating that the age and the assay performed are not responsible for the finding [16].

Due to its efficacy, in recent years, propranolol has been widely used in IH therapy, although its mechanism of action has not yet been understood. One hypothesis is that propranolol acts on hemangioma endothelial cells influencing vasoconstriction and blocking angiogenic factors [31]. Very recently, it has been documented in vitro that propranolol favors and accelerates the involution phase of the disease, inducing adipogenesis in CD133+ hemangioma cells [32]. In this study, we investigated ex vivo the influence of propranolol on cMSCs of patients with IH. A gradual decrease of cMSC frequency was observed after one-year therapy, while the therapy did not influence the cMSCs proliferative capacity or the number of passages needed to reach their senescence status.

Interestingly, the differentiation of cMSCs into adipocytes, significantly defective in patients with IH at T0, was improved in patients receiving therapy for 9–12 months, as suggested by the up-regulated levels of both *PPARγ* and *C/EBPα*. These data are partially in keeping with those of studies reporting an accelerated adipogenesis followed by a rapid apoptosis-independent cell death [33] possibly caused by a high propranolol concentration in in vitro cultures [12,34]. Moreover, the adipogenic gene expression induced by propranolol was investigated, indicating an increase of *PPARγ* and a failure in the induction of *C/EBPα* by *C/EBPβ* and *C/EBPδ* [33,34]. Taken together, these data indicate that propranolol added in vitro allows a deregulated adipogenesis. In our hands, no evidence of death has been recorded during the increased adipogenic differentiation of cMSCs from patients who received propranolol, suggesting that the drug plasma concentration has no influence on specific adipocyte survival factors, as described by Wong et al. in a study performed using both hemangioma stem cells and BM-derived MSCs [34]. Regarding the adipogenic gene expression, we found an increase of both *PPARγ* and *C/EBPα* in cMSCs of patients receiving propranolol, suggesting a physiological adipocyte differentiation. In addition, we noted the appearance of an increased number of lipid droplets in cMSCs of patients who received propranolol cultured in the absence of stimulating factors. Similarly, Wong et al. described that, even in the absence of adipogenic stimuli, propranolol added in vitro induced the expression of adipogenic genes in hemangioma stem cells [34].

We report an additional finding associated to therapy with propranolol; cMSCs from patients not yet receiving therapy were able to support the formation of tube-like structures in co-cultures with healthy ECFCs, while no tube-like structures characterized the co-cultures of cMSCs from patients at the end of the therapy. The lack of tube-like structures was evidenced also in the co-culture of healthy BM MSCs and healthy ECFCs. These data, that set cMSCs as a target of propranolol, are in keeping with studies documenting an inhibitory role of propranolol on angiogenesis: ex vivo, by significantly declining the circulating VEGF levels of patients after 3 months of therapy [35], and in vitro, by tube formation inhibition of HUVECs in the presence of VEGF [36].

Data from gene expression indicated that cMSCs and BM MSCs were distinct entities from ECFCs and HUVECs. The analysis restricted to cMSCs from patients with IH at T0 versus CTRLs showed the increased expression of the pro-inflammatory *IL1β* gene and the decreased expression of *ESM1*, a gene having a role either in containing inflammation or in favoring angiogenesis [37]. The role of inflammation in supporting angiogenic processes is suggested by the increased expression of *F3* gene [38]. The reduced expression of *F3* gene in T12 compared to T0 cMSCs speaks in favor of a propranolol role in dampening a potential proangiogenic activity of cMSCs in IH. In addition, the upregulation of *EDN1* gene expression at the end of therapy may reflect the capacity of cMSCs to support also mechanisms leading to tissue fibrosis [39]. The downregulation of *TGFα* gene expression in cMSCs of IH at T0 and its upregulation after propranolol therapy are not consensual with their proangiogenic activity. However, very recently, it has been reported both an inflammatory environment in IH with multiple lesions [40] and a role of inflammation in defining the paracrine profile of MSCs, possibly altering the expression of angiogenic cytokines such as TGFα, in favor of a major reparative capacity [41]. In addition, propranolol and its isomers have been described in in vitro studies to promote the internalization of the MSC membrane receptors binding either EGF or TGFα; a similar mechanism may alter in vivo the *TGFα* gene expression [42]. Of note, cMSCs of patients at the end of therapy with propranolol showed an expression profile not different from that of cMSCs from CTRLs.

The number of patients assessed and the very limited availability of healthy subjects comparable in age represent the main limitations of this study, since they prevented us from performing in some cases a sufficient number of experiments to reach statistical significance. This reflects both the understandable poor compliance of the parents in giving consent for study participation and the actual difficulty in obtaining the blood samples from neonates.

## 5. Conclusions

For the first time we have reported that MSCs are detectable in the PB of patients with IH and favor the angiogenic activity in vitro. The propranolol therapy downregulates this activity and induces cMSC differentiation in adipocytes, possibly counteracting the tumor growth and/or changing of the nature of the tumor tissue. Other factors, such as the aging of the patients or the physiological tendency to involution of the IH, could be involved in the changes of cMSC gene expression observed after therapy.

## Figures and Tables

**Figure 1 cells-13-00254-f001:**
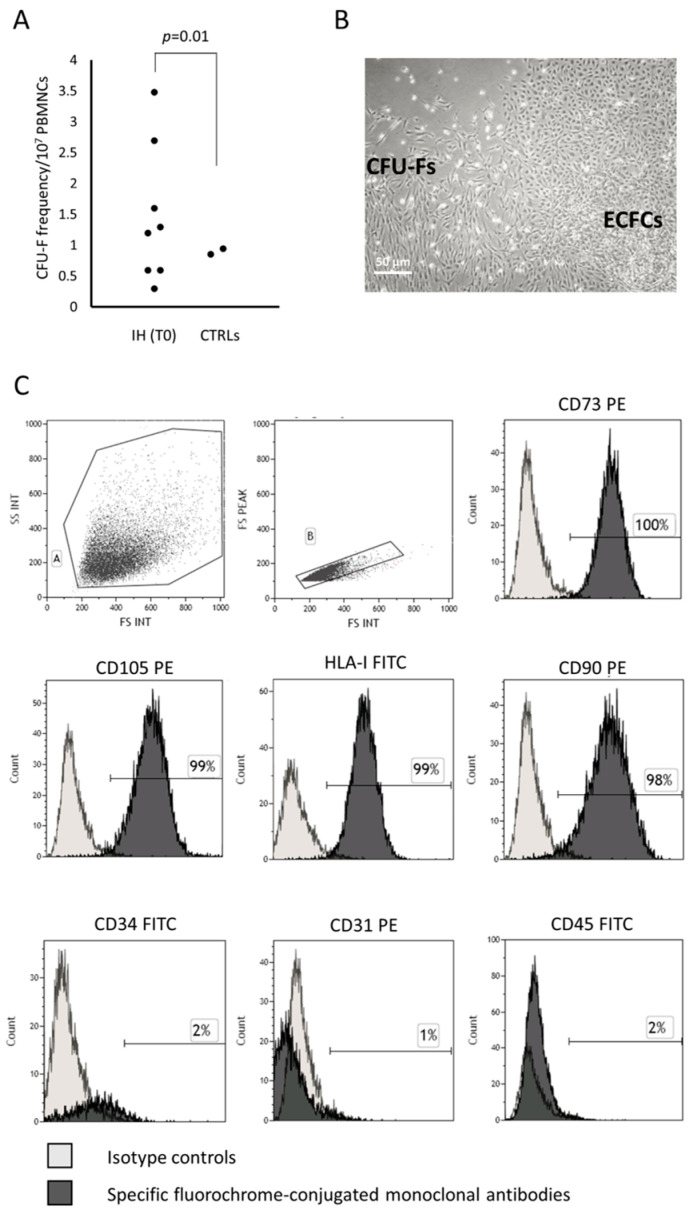
CFU-Fs co-existing with ECFCs in in vitro cultures show the MSC phenotype. Colony forming unit-fibroblast (CFU-F) frequency in peripheral blood mononuclear cells (PBMNCs) of patients with infantile hemangioma (IH) at the onset (T0) and in healthy subjects comparable in age (CTRLs) is shown (**panel A**). Photomicrograph of CFU-Fs and endothelial colony forming cells (ECFCs) observed in plate where PBMNCs from one representative patient with IH at T0 has been cultured (original magnification 10×) (**panel B**). The histograms show the CFU-F-derived cell phenotype (**panel C**). The percentage of positive cells in gate A and B was calculated, subtracting the value of the appropriate isotype controls.

**Figure 2 cells-13-00254-f002:**
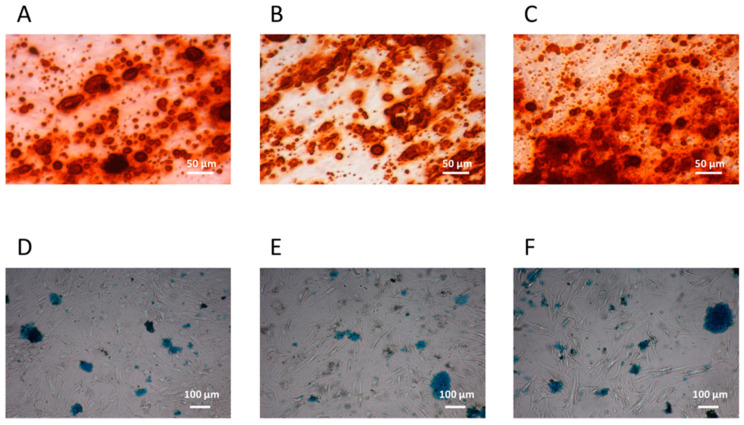
cMSC osteogenic and chondrogenic differentiation. Osteogenic and chondrogenic differentiation capacity of circulating mesenchymal stromal cells (cMSCs) from a representative patient with IH at T0 (**panels A** and **D**, respectively) and from a CTRL (**panels B** and **E**, respectively); the bone marrow (BM) MSC osteogenic (**panel C**) and chondrogenic (**panel F**) differentiation capacity are also shown. The osteoblasts differentiation is demonstrated by the histological detection of calcium depositions positive for Alizarin Red S (magnification 10×); the chondrocyte differentiation is demonstrated by the histological detection of Alcian blue 8GX deposition (magnification 4×).

**Figure 3 cells-13-00254-f003:**
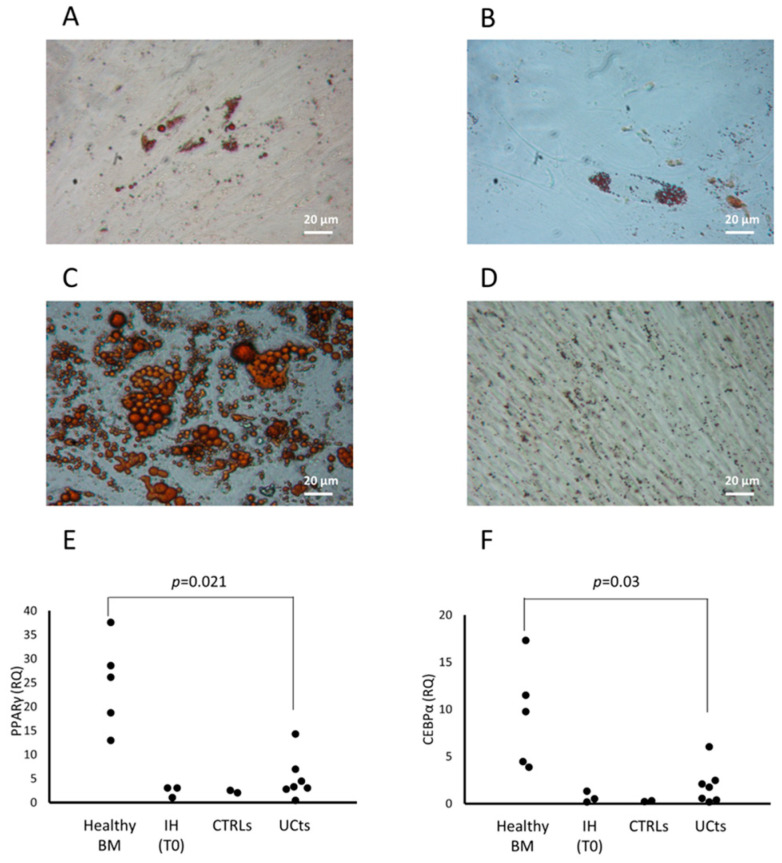
cMSC adipocyte differentiation. cMSC differentiation into adipocytes is revealed by the formation of lipid droplets stained with Oil Red O (magnification 20×) in one representative patient with IH at T0 (**panel A**), and in the cMSCs of a CTRL (**panel B**). The differentiation into adipocytes of MSCs from a representative BM (**panel C**) or from umbilical cord tissue (UCt) (**panel D**) is shown. The expression of the peroxisome proliferator-activated receptor gamma (PPARγ) transcription factor (**panel E**) and of the CCAAT/enhancer-binding protein alpha (C/EBPα) (**panel F**) are assessed in BM- and UCt-MSCs, in cMSCs from patients with IH at T0 and in CTRLs. Results are expressed as relative quantification (RQ) using Hypoxanthine phosphoribosyltransferase 1 (HPRT-1) as internal control.

**Figure 4 cells-13-00254-f004:**
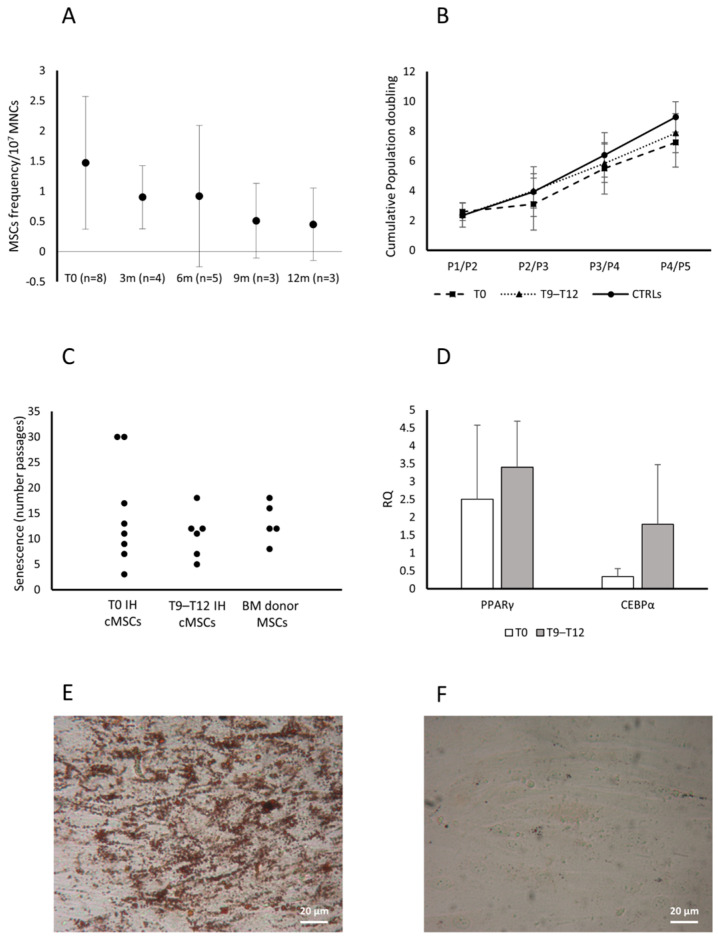
Characterization of cMSCs in patients with IH receiving propranolol. CFU-F frequency in patients with IH assessed at T0 and while receiving propranolol at 3, 6, 9 and 12 months. Data are expressed as number of colonies in 10^7^ MNCs (**panel A**). Cumulative population doublings (cPDs) from passage P1 to P5 of cMSCs isolated from patients with IH at T0 (squares), after 9–12 months of therapy with propranolol (T9–T12, triangles) and CTRLs (circles) (**panel B**). Passages at which cMSCs of patients with IH at T0 or at T9–T12 and MSCs from healthy BMs entered senescence are reported (**panel C**). Expression levels of both PPARγ and CEBPα in adipocytes from cMSCs of patients with IH at T0 (white columns) and at T9–T12 (gray columns) (**panel D**); results are expressed as RQ using HPRT-1 as internal control. Data in panels A, B and D are shown as mean (±SD). Lipid droplets stained with Oil Red O (magnification 20×) in cMSCs cultured in medium without the stimuli that induce the adipogenic differentiation from two representative patients with IH, one after 9–12 month therapy with propranolol (**panel E**) and one at T0 (**panel F**).

**Figure 5 cells-13-00254-f005:**
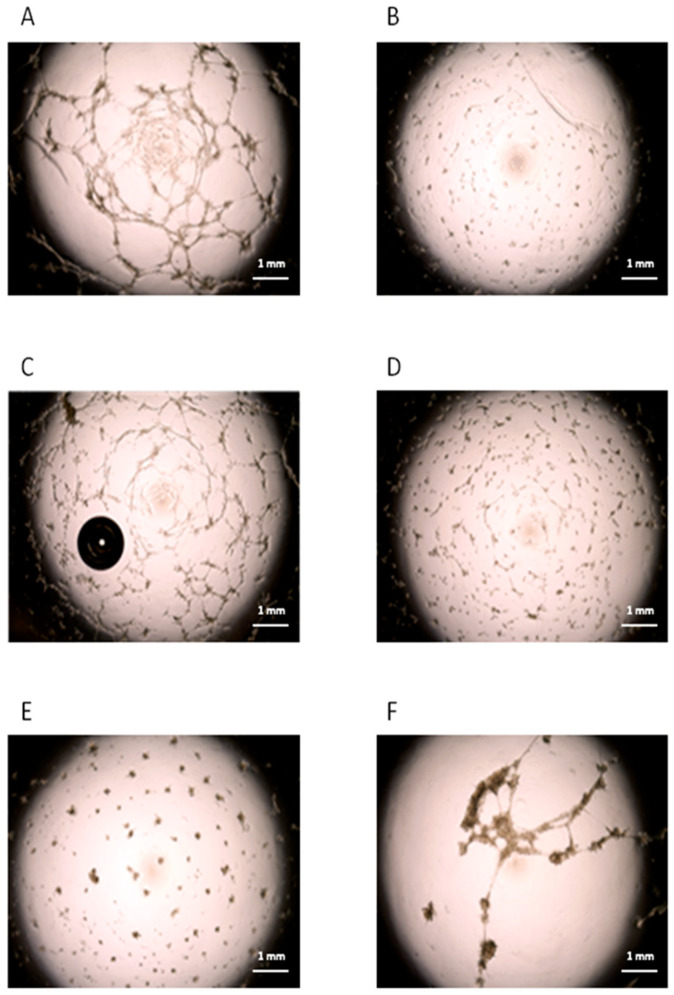
Angiogenic role of cMSCs from patients with IH. Tube-like structures obtained by in vitro cultures of ECFCs from a healthy subject in standard (**panel A**) or suboptimal conditions (**panel B**). ECFCs from the healthy subject were cultured in suboptimal conditions in the presence of cMSCs (ratio 5:1) from patients with IH at T0 (**panel C**), or from CTRLs (**panel D**). MSCs from BM donors were also co-cultured with ECFCs (**panel E**). ECFCs from the CTRL were cultured in suboptimal conditions in the presence of cMSCs (ratio 5:1) from patients with IH after 12 months of therapy with propranolol (**panel F**). Each panel in the figure refers to one representative experiment.

**Figure 6 cells-13-00254-f006:**
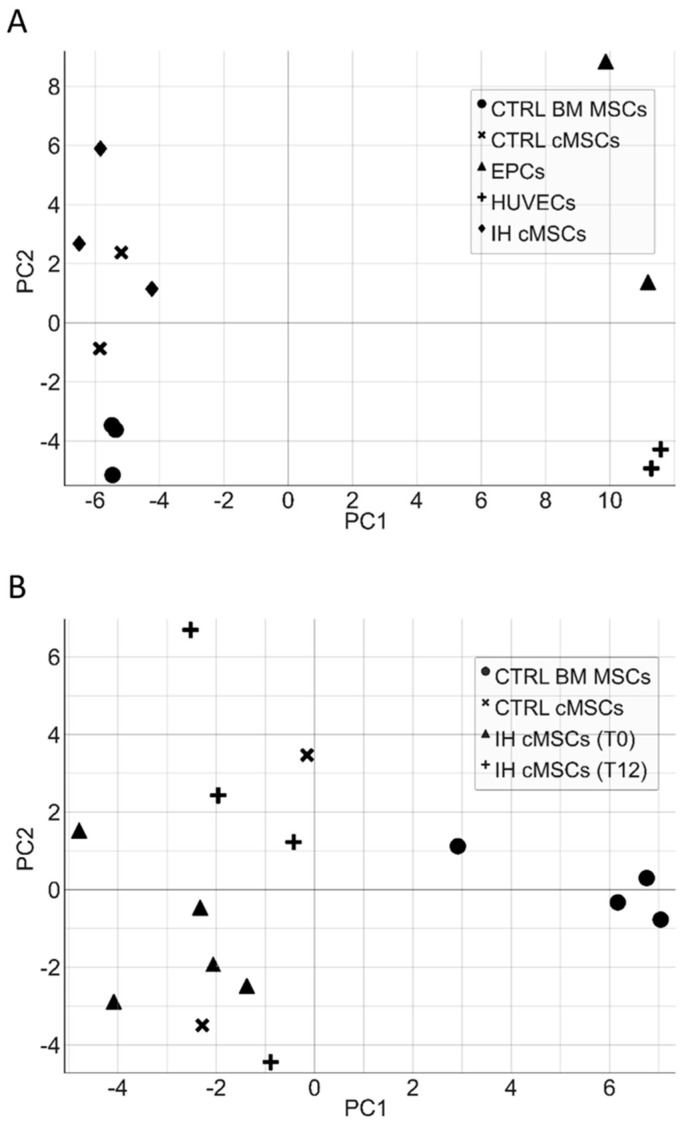
Principal component analysis of endothelial cells and MSCs. The principal component analysis (PCA) of a different subset of endothelial cells (ECFCs, HUVECs) and cMSCs from IH patients shows a clear separation of the gene expression between cells of the endothelial compartment and MSCs (**panel A**). Similarly, the PCA of cMSCs from patients and CTRLs versus BM MSCs highlights a gene expression difference between MSCs derived from PB and those derived from BM (**panel B**).

**Figure 7 cells-13-00254-f007:**
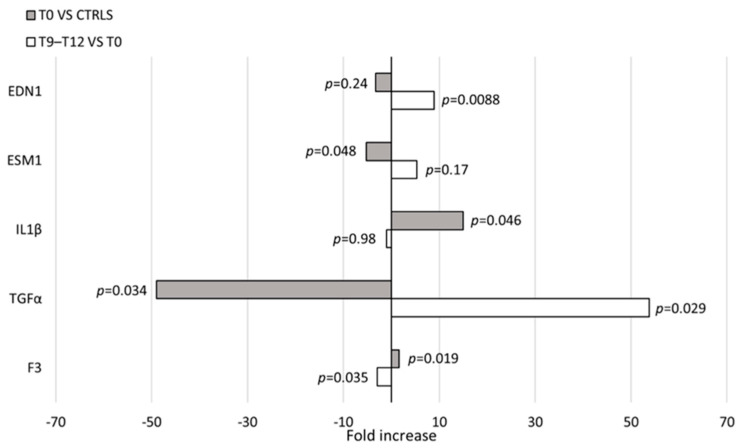
Gene expression in cMSCs. Evaluation of gene expression in cMSCs of patients with IH at T0 (*n* = 5) and at T9–T12 (*n* = 4). *p* values between patients at T0 and CTRLs (*n* = 2) (gray bars) or patients at T0 and at T9–T12 (white bars) are shown.

## Data Availability

The raw data supporting the conclusions of this article will be made available by the authors on request.

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
