# Peer review of "Circulating Mesenchymal Stromal Cells in Patients with Infantile Hemangioma: Evaluation of Their Functional Capacity and Gene Expression Profile"

_cells, 2024, doi:10.3390/cells13030254_

Round 1

Reviewer 1 Report

Comments and Suggestions for Authors

Starting from previous work by the same Authors, the presents study has been aimed to characterize the circulating colony forming units-fibroblasts (CFU-Fs) that are present in children affected by infantile hemangioma (IH). The results shown herein indicate that: i) circulating CFU-Fs are detectable in IH patients at numbers higher than those present in healthy controls; ii) the phenotype of circulating CFU-Fs is that of mesenchymal stromal cells (MSCs); iii) CFU-Fs obtained from IH patients differ from MSCs of healthy controls in the expression of inflammatory and angiogenic cytokines; iv) at variance with control MSCs, the CFU-Fs of IH patients promote angiogenesis in vitro; v) when IH patients are treated for one year with the beta-adrenergic receptor antagonist propanolol, the number and phenotypic features of their CFU-Fs become similar to those of MSCs obtained from healthy controls.  

Taken together, these findings provide information to better understand IH pathogenesis, as well as to devise new tools for the diagnosis and to individuate new targets for the therapy of IH. Although the statistical significance of the results is somehow compromised by the small number of samples here analyzed, the validity of this study is guaranteed by the accuracy of its experimental design.

Comments on the Quality of English Language

In my opinion, the quality of English language is good.

Author Response

We thank the reviewer for the comments.

Reviewer 2 Report

Comments and Suggestions for Authors

Abbá et al.  have analyzed the properties of CFU-Fs present in the blood of patients suffering infantile hemangiomas, prior to and during long term treatment with propranol. They observe that these cells present phenotypic characteristics of MSC, inicluding a capacity for pluripotent differentiation. These CFU-F are more often observed in IH patients than in CTRL not suffering form IH. They also present unique angiogenetic properties as compared to these latter and to bone-marrow-derived MSC. These angiogenic properties seem to disappear following long-term propranolol treatment. Altogether, the authors provide a very welcome and valuable insight on the (yet unknown) mechanism of action of propranolol as a treatment of IH.

Minor Comments:

Statistics:

1.     results are reported as means +/- SE, which is not adequate in reporting series of individual samples- Please change to SD;

2.     the analysis of the frequencies of PB CFU-F between CTRL and IH patients (8/18 vs. 2/24) should be performed using Fisher’s exact test- this provides a P value of 0.0104, and thus does not alter the conclusion of the authors, but is more appropriate (please correct as well the M&M section;

CFU-F differentiation

It is unclear how the authors reach the conclusion that the CFU-F can be defined as c-MSCs, given that that their best adipocyte differentiation capacity at To is similar (and even inferior, based on Fig 3) to that of umbilic cord tissue cells, but significantly inferior to that of true cMSC; These cells however definitely present the possibility to differentiate in bone and cartilage, and thus present some features of MSC indeed. Please nuance this statement, and I’d propose to refer to them as ‘CFU-F rather than MSC further in the manuscript.

Effect of propranolol treatment on the adipocytic differentiation of CFU-F

The observed differences in the adipocyte differentiation of CFU-F is interesting but may not be specific to the CFU-F found in IH patients- How does propranolol treatment (in vitro) affect the adipocyte differentiation of To CFU-F form patients, UCt and CFU-C obtained form CTRLs?

Author Response

Minor comments

Statistics:

1.Results are reported as means +/- SE, which is not adequate in reporting series of individual samples- Please change to SD

In the revised version of the manuscript the results are reported as mean±SD

2.The analysis of the frequencies of PB CFU-F between CTRL and IH patients (8/18 vs. 2/24) should be performed using Fisher’s exact test- this provides a P value of 0.0104, and thus does not alter the conclusion of the authors, but is more appropriate (please correct as well the M&M section;

As suggested, the analysis of the frequencies of PB CFU-Fs between CTRLs and IH patients (8/18 vs.2/24) has been performed using Fisher’s exact test. This has been reported also in the M&M section.

CFU-F differentiation

It is unclear how the authors reach the conclusion that the CFU-F can be defined as c-MSCs, given that that their best adipocyte differentiation capacity at To is similar (and even inferior, based on Fig 3) to that of umbilic cord tissue cells, but significantly inferior to that of true cMSC; These cells however definitely present the possibility to differentiate in bone and cartilage, and thus present some features of MSC indeed. Please nuance this statement, and I’d propose to refer to them as ‘CFU-F rather than MSC further in the manuscript.

We agree with the reviewer regarding the fact that the best differentiation in adipocyte obtained from cMSCs of patients with IH at T0 is inferior to that of umbilical cord tissue cells and BM MSCs. However, it has been reported (i.e. Mastrolia I et al Stem cells Translational Medicine 2019;8:1135-1148) that MSC population, isolated from a variety of sources, has peculiar degrees of capacity to differentiate,  as shown by comparative analysis. The differences are based on the tissue of origin as well as donor related variability (gender, age, disease) or isolation procedure. These evidences allow to consider as  cMSCs CFU-F cells that have spindle shape morphology, the MSC phenotype, self renewal capacity and differentiation capacity comparable or lower than that of BM MSCs. We emphasized this at the end of the paragraph 3.1.2

Effect of propranolol treatment on the adipocytic differentiation of CFU-F

The observed differences in the adipocyte differentiation of CFU-F is interesting but may not be specific to the CFU-F found in IH patients- How does propranolol treatment (in vitro) affect the adipocyte differentiation of To CFU-F form patients, UCt and CFU-C obtained form CTRLs?

We understand your point. We performed a number of experiments using in vitro cultures of cMSCs from patients with IH at the onset in presence/absence of a wide range of propranolol concentrations (including the maximunm/minimal concentrations reached in plasma following assumption of the drug). As described with hemangioma derived stem cells, we observed a rapid death of cMSCs, independently from the drug concentration. We did not perform experiments with UCt MSCs or CTRL cMSCs.

As discussed in the manuscript, there are a number of published studies reporting an accelerated adipogenesis followed by a rapid apoptosis-independent death of hemangioma derived stem cells [1] possibly caused by a high propranolol concentration in in vitro cultures [2,3]. Moreover, the adipogenic gene expression induced by propranolol was investigated, indicating an increase of PPARγ and a failure in the induction of C/EBPα by C/EBPb and C/EBPg [1,3]. These data indicate that propranolol added in vitro allows a deregulated adipogenesis. No evidence of death has been recorded during the increased adipogenic differentiation of cMSCs from  patients who received propranolol, suggesting that the drug plasma concentration has no  influence on specific adipocyte survival factors, as described by Wong et al in a study performed using both hemangioma stem cells and BM-derived MSCs. Regarding the adipogenic gene expression, we found an increase of both PPARγ and C/EBPα in cMSCs of patients receiving propranolol suggesting a physiological adipocyte differentiation. In  addition, we noted the appearance of an increased number of lipid droplets in cMSCs of  patients who received propranolol cultured in absence of stimulating factors. Similarly, Wong et al described that even in the absence of adipogenic stimuli, propranolol added  in vitro induced the expression of adipogenic genes in hemangioma stem cells. No studies have been published using cMSCs.

1.England, R.W.; Hardy, K.L.; Kitajewski, A.M.; Wong, A.; Kitajewski, J.K.; Shawber, C.J.; Wu, J.K. Propranolol promotes 562 accelerated and dysregulated adipogenesis in hemangioma stem cells. Ann Plast Surg 2014, 73 Suppl 1, S119-124

2.Li, H.H.; Lou, Y.; Zhang, R.R.; Xie, J.; Cao, D.S. Propranolol Accelerats Hemangioma Stem Cell Transformation Into Adipo-501 cyte. Ann Plast Surg 2019, 83, e5-e13

3.Wong, A.; Hardy, K.L.; Kitajewski, A.M.; Shawber, C.J.; Kitajewski, J.K.; Wu, J.K. Propranolol accelerates adipogenesis in 565 hemangioma stem cells and causes apoptosis of hemangioma endothelial cells. Plast Reconstr Surg 2012, 130, 1012-1021, 566

Reviewer 3 Report

Comments and Suggestions for Authors

The manuscript explores a fascinating topic, aiming to characterize circulating mesenchymal stromal cells in patients with infantile hemangioma by isolating the cells from peripheral blood. However, substantial improvements are required to enhance clarity and presentation. Major concerns involve the need for a comprehensive rewrite and reformatting of the manuscript, particularly in the context of Figure formatting, which lacks essential information such as scale bars, labeling, and statistical details.

Major Comments:

Figure Formatting:

All figures need a complete reformatting. Crucial missing information includes scale bars, labeling, and statistical annotations. The figures should clearly indicate controls and tested samples, incorporate color-coded symbols for antibodies, and provide a gating strategy to depict the percentage of the population. Additionally, avoid separating Figure 1D from Figures A, B, and C by using bordering, and ensure a cohesive arrangement for improved clarity.

Specific Minor Comments:

Abstract (Line 27): Introduce the field briefly in one sentence to provide context for the study.

Lines 59-77: Trim excessive self-citations and focus on explaining the research gap and the rationale for conducting the current study.

Line 107: should write in full for the first time: Human Ucts

Line 115: D-MEM (should mention the manufacturer like other reagents)

Line 133: To induce adipogenic differentiation medium … Should separete the Subject: To induce adipogenic differentiation, the medium

Line 152: DD cq method: should insert an symbol of delta Δ or full words

Line 162-163: “cDNA 161 was quantified spectrophotometrically using NanoDrop” : please check: usually not check cDNA with nanodrop

What is the different between 2.6 and 2.7, both describe mRNA gene expression analysis

Line 181: should verify the information: standard error (SE) is SEM or SD

Line 199: “ The CFU-Fs were expanded and characterized according to the criteria defined by 199 the International Society for Cellular Therapy for MSCs [23].” Should shortly describe here

Figure 1 (B, C, D):

Include a scale bar in Figures B and C. In Figure D, employ symbols or coloring to differentiate between control and tested samples. Indicate the fluorophores beside antibody names, and incorporate a gating strategy for percentage representation.

Line 201:Avoid referencing images from previous papers; provide the necessary details directly.

Lines 207-210:Elaborate on the results rather than merely stating what is depicted in the figures.

Line 216:Clarify which information the author intends to convey or compare with references [20, 24].

Figure 4D:

Incorporate statistical analysis details directly on the plot.

Part 3.4: Supplement PCA with a bar graph to illustrate the expression levels of the genes of interest.

Tables 2 and 3:

Consider moving Table 2 to the supplementary section and replacing Table 3 with a graph for improved visualization.

In summary, while the manuscript delves into an intriguing subject, substantial revisions are necessary to enhance its clarity and presentation. The suggested improvements aim to address major concerns related to figure formatting and specific minor issues throughout the manuscript.

Comments on the Quality of English Language

Have to improve 

Author Response

Figure formatting

All figures need a complete reformatting. Crucial missing information includes scale bars, labeling, and statistical annotations. The figures should clearly indicate controls and tested samples, incorporate color-coded symbols for antibodies, and provide a gating strategy to depict the percentage of the population. Additionally, avoid separating Figure 1D from Figures A, B, and C by using bordering, and ensure a cohesive arrangement for improved clarity.

As suggested by the reviewer we reformatted all figures adding scale bars (figures 1, 2, 3 and 4) and statistical annotations. In addition, in figure 1 we added antibody labels, statistical annotations and a dot plot showing the gating strategy used to define the CFU-F population. Moreover, we avoided separating plots using bordering and removed the previously published picture (panel B, Patient 1) as suggested. All the figure panels were then arranged aiming to improve the clarity of our data.

Specific minor comments

Abstract (Line 27): Introduce the field briefly in one sentence to provide context for the study.

We introduced the field briefly to provide the context for the study

Lines 59-77: Trim excessive self-citations and focus on explaining the research gap and the rationale for conducting the current study.

We modified the introduction focusing on explaining the research gap and the rationale for conducting the current study

Line 107: should write in full for the first time: Human Ucts

Human UCts was written in full in the last line of the introduction, where it appeared for the first time

Line 115: D-MEM (should mention the manufacturer like other reagents)

We added the information in M&M section

Line 133: To induce adipogenic differentiation medium … Should separete the Subject: To induce adipogenic differentiation, the medium

According with the reviewer suggestion, we modified the sentence

Line 152: DD cq method: should insert an symbol of delta Δ or full words

We inserted the symbols

Line 162-163: “cDNA 161 was quantified spectrophotometrically using NanoDrop” : please check: usually not check cDNA with nanodrop

We thank the reviewer for this observation. Actually, the nanodrop was used to quantify the RNA

What is the different between 2.6 and 2.7, both describe mRNA gene expression analysis

We agree with the reviewer regarding the fact that both paragraph 2.6 and 2.7 describe mRNA gene expression analysis. However, the procedures described in the two paragraphs did not completely overlap and the reagents used were different; it seemed more correct to describe the procedures in separated paragraphs

Line 181: should verify the information: standard error (SE) is SEM or SD

In the revised version of the manuscript we modified the panel A, B and D of figure 4 showing the standard deviation values as requested also by reviewer 2

Line 199: “ The CFU-Fs were expanded and characterized according to the criteria defined by 199 the International Society for Cellular Therapy for MSCs [23].” Should shortly describe here

As suggested, we shortly described the published criteria used by the scientific community to define MSCs

Figure 1 (B, C, D): Include a scale bar in Figures B and C. In Figure D, employ symbols or coloring to differentiate between control and tested samples. Indicate the fluorophores beside antibody names, and incorporate a gating strategy for percentage representation.

As explained above, we modified Figure 1 according to the reviewer’s suggestions

Line 201:Avoid referencing images from previous papers; provide the necessary details directly.

According with the reviewer’s suggestion, we eliminated the referencing image from Figure 1

Lines 207-210:Elaborate on the results rather than merely stating what is depicted in the figures.

We shorted and modified this part of the results

Line 216:Clarify which information the author intends to convey or compare with references [20, 24].

These references refer to previous studies showing that umbilical cord tissue derived MSCs had a defective differentiation in adipocytes with respect to BM MSCs indicating that our current finding is consistent

Figure 4D: Incorporate statistical analysis details directly on the plot.

The statistical analysis performed with data shown in figure 4D was not statistically significant; for this reason we did not incorporate the p values on the plot

Part 3.4: Supplement PCA with a bar graph to illustrate the expression levels of the genes of interest.

We show in a bar graph the expression levels of the 48 genes, selected according to their differential expression in the initial analysis of 147 genes, in the different cell populations (Figure S1 and S2 in supplementary material)

Tables 2 and 3: Consider moving Table 2 to the supplementary section and replacing Table 3 with a graph for improved visualization.

As suggested, we moved Table 2 to the supplementary section (Table S1)

In the revised manuscript, Table 3 has been replaced by a bar graph (Figure 7)

Round 2

Reviewer 3 Report

Comments and Suggestions for Authors

This manuscript reports on an interesting topic, characterizing the circulating mesenchymal stromal cells in patients with infantile hemangioma by isolating the cells from peripheral blood (PB). The manuscript has been improved compared to the first version. Below are specific minor comments for the current version of the manuscript:

Abstract:

The missing field and the rationale for the current study are not clearly explained.

Briefly mention the key methods used in the study.

Rewrite the results section with key findings relevant to this study, linking them with the conclusion.

Line 90:

Range 1-30: Are these months?

Line 129:

[16]: Why is there a reference at this part?

Line 150-151:

"qPCR methods to test (i) RNA integrity, (ii) genomic DNA contamination, and (iii) presence of PCR inhibitors": Clarify the purpose of these tests, as the results only show gene expression.

Line 157:

"10ng of total RNA": Please double-check, as this seems to be a very small amount. Can the kit detect this amount of cDNA?

Figure 1:

Increase the text size of all panels.

For panel C, include proper gating of the main population. Consider adding a plot showing singlet gating. Ensure all other panels display fluorescence intensity on the X-axis.

For the CD45 panel, differentiate isotype and CD45 with different colors.

Add scale bars to microscope images in all figures. Reduce blank space between panels and increase text size.

Line 252:

"The histograms show the CFU-F derived cell phenotype (panel C)": Reword this part to provide more detailed information.

Figure 5:

Add a scale bar.

Figure 6:

Reduce the size of the PCA plot but increase the text size. Consider using different colors to distinguish groups. The current version is difficult to interpret, especially for panel A: IH cMSCs group.

Line 435:

"Data from gene expression profiling": Avoid using "profiling" when the study involves testing only several genes. Profiling is typically done with arrays or RNA seq for multiple genes.

Comments on the Quality of English Language

Minor editing of English language required

Author Response

 Abstract:

The missing field and the rationale for the current study are not clearly explained.

Briefly mention the key methods used in the study.

Rewrite the results section with key findings relevant to this study, linking them with the conclusion.

According with the reviewer’s suggestions the abstract has been rewritten

Line 90:

Range 1-30: Are these months?

Yes, as indicated for the median value: (median age 7 months, range 1-30)

Line 129:

[16]: Why is there a reference at this part?

We agree with the reviewer, the ref is not useful and therefore it has been eliminated

Line 150-151:

"qPCR methods to test (i) RNA integrity, (ii) genomic DNA contamination, and (iii) presence of PCR inhibitors": Clarify the purpose of these tests, as the results only show gene expression.

We agree with the reviewer, we evaluated only the gene expression. This part has been eliminated.

Line 157:

"10ng of total RNA": Please double-check, as this seems to be a very small amount. Can the kit detect this amount of cDNA?

We thank the reviewer for this observation, actually the amount of total RNA was 100ng. We modified the value in the manuscript

Figure 1:

Increase the text size of all panels.

For panel C, include proper gating of the main population. Consider adding a plot showing singlet gating. Ensure all other panels display fluorescence intensity on the X-axis.

As suggested by the reviewer, we modified Figure 1. We added a plot showing the singlet gating. Regarding the display of fluorescence intensity on the X-axis, we used the Kaluza software, which does  not show the fluorescence intensity on the X-axis when you select the “merge” option. We added these details in paragraph 2.3

For the CD45 panel, differentiate isotype and CD45 with different colors.

We understand your point; in the revised version of the manuscript we added in each histogram the percentage of positive cells and this should help the readers.

Add scale bars to microscope images in all figures. Reduce blank space between panels and increase text size.

As suggested by the reviewer, we added the scale bars in the microscope images of all the figures. Where possible, we reduced the blank spaces.

Line 252:

"The histograms show the CFU-F derived cell phenotype (panel C)": Reword this part to provide more detailed information.

We added a detailed description in paragraph 2.3 and specified the calculation of positive cells in the figure legend.

Figure 5:

Add a scale bar.

The scale bars were added

Figure 6:

Reduce the size of the PCA plot but increase the text size. Consider using different colors to distinguish groups. The current version is difficult to interpret, especially for panel A: IH cMSCs group.

As suggested by the reviewer, we reduced the size of PCA plot and  increased the text size. The groups are now represented by bigger and darker symbols that make the current version of the figure more readable

Line 435:

"Data from gene expression profiling": Avoid using "profiling" when the study involves testing only several genes. Profiling is typically done with arrays or RNA seq for multiple genes.

As suggested, we avoided the use of “profiling” in the result and discussion sections